# XLUMINA:
# An Auto-differentiating Discovery Framework for Super-Resolution Microscopy

**Carla Rodríguez**
Max Planck Institute for the Science of Light
Erlangen, Germany
`carla.rodriguez@mpl.mpg.de`

**Sören Arlt**
Max Planck Institute for the Science of Light
Erlangen, Germany
`soeren.arlt@mpl.mpg.de`

**Leonhard Möckl**
Max Planck Institute for the Science of Light
Erlangen, Germany
`leonhard.moeckl@mpl.mpg.de`

**Mario Krenn**
Max Planck Institute for the Science of Light
Erlangen, Germany
`mario.krenn@mpl.mpg.de`

## Abstract

In this work we introduce XLUMINA, an original computational framework designed for the discovery of novel optical hardware in super-resolution microscopy. Our framework offers auto-differentiation capabilities, allowing for the fast and efficient simulation and automated design of entirely new optical setups from scratch. We showcase its potential by rediscovering three foundational experiments, each one covering different areas in optics: an optical telescope, STED microscopy and the focusing beyond the diffraction limit of a radially polarized light beam. Intriguingly, for this last experiment, the machine found an alternative solution following the same physical principle exploited for breaking the diffraction limit. With XLUMINA, can we go beyond simple optimization and calibration of known experimental setups, opening the door to potentially uncovering new microscopy concepts within the vast landscape of experimental possibilities.

## 1 Introduction

The space of all possible experimental optical configurations is enormous. For example, if we consider experiments that consist of just 10 optical elements, chosen from 5 different components (such as lasers, lenses, phase shifters, beam splitter and cameras), we already get 10 million possible discrete arrangements. The experimental topology will further increase this number greatly. Finally, each of these optical components can have tunable parameters (such as lenses' focal lengths, laser power or splitting ratios of beam splitters) which lead to additional high-dimensional continuous parameter space for each of the previously mentioned discrete possibilities. This vast search space contains all experimental designs possible, including those with exceptional properties. So far, researchers have been exploring this space of possibilities guided by experience, intuition and creativity – and have uncovered countless exciting experimental configurations and technologies. But due to the complexity of this space, it might be that some powerful concepts and techniques have not been discovered so far, and might never be with a human-driven direct design approach. This is where AI-based exploration techniques could provide enormous benefit, by exploring the space in a fast, unbiased way [1, 2].

Optical microscopes in today's sense were invented 300 years ago by Antonj van Leeuwenhoek [3]. Since then, few techniques used in the sciences have seen a similarly rapid development and impact

NeurIPS 2023 AI for Science Workshop.

on diverse fields, ranging from material sciences all the way to medicine [4–7]. Arguably, optical microscopy is currently most widely used in biological sciences, where precise labeling of imaging targets enables fluorescence microscopy with exquisite sensitivity and specificity [8, 9]. In the past two decades, several breakthroughs have broadened the scope of optical microscopy in this area even further. Among them, the discovery of super-resolution (SR) methods, which circumvent the classical diffraction limit of light, stand out in particular. Examples for versatile and powerful SR techniques are STED [10], PALM/F-PALM [11, 12], (d)STORM [13, 14], SIM [15], and MINFLUX [16], with considerable impact in biology [17–19], chemistry [20] and material sciences [21] for example. These microscopy techniques were designed through the ingenuity and creativity of human researchers. Crucially, the motivation of our work goes far beyond small-scale optimization of already known optical techniques. Rather, this work sets out to discover novel, experimentally viable concepts for advanced optical microscopy that are at-present entirely untapped.

Fundamentally, the simulator is the heart of digital discovery efforts. It translates an experimental design (one point in the vast space of possible designs) to a physical output. The physical output, such as a detector or a camera, can then be used in an objective function to describe the desired design goal. The simulator can either be called directly by gradient-based optimization techniques, or it can be used for generating the training data for deep-learning-based surrogate models. A simulator that can be used for automated design and discovery of new experimental strategies must be (1) fast, (2) reliable, and (3) general. In our manuscript, we present a simulator that fulfills precisely the aforementioned requirements for advanced microscopy. We leverage its scope with a specific focus on the area of super-resolution microscopy, which is a set of techniques that has revolutionized biological and biomedical research over the past decade, highlighted by the 2014 Chemistry Nobel Prize [22].

We introduce XLUMINA, an efficient framework with auto-differentiation capabilities [23] for the ultimate goal of discovering new optical design principles. We demonstrate our approach on three foundational optical layouts: a telescope version, the polarization-based beam shaping as used in STED (stimulated emission depletion) microscopy [10] and the sharp focus of a radially polarized light beam [24]. The obtained results not only yielded a rediscovery of these foundational configurations but also unveiled novel solutions following the same underlying physical principle present in these experiments. Crucially, the motivation of our work goes far beyond small-scale optimization of already known optical techniques. Rather, the future application of XLUMINA is the AI-driven discovery of completely novel physical concepts for advanced optical microscopy.

## 1.1 Previous work

**Optimization in microscopy**    Our approach is radically different from previous strategies that employ AI for data-driven design of single optical elements [25, 26] or data analysis in microscopy, e.g. denoising, contrast enhancement or point-spread-function (PSF) engineering [27]. While these techniques are influential, they are not meant to change the principle of the experimental approach or the optical layout itself. In contrast, XLUMINA is equipped with tools for simulate, optimize and automatically design new optical setups and concepts from scratch.

**Discovery in quantum optics**    Numerous works have recently shown how to automatically design new quantum experiments with advanced computational methods [28–31], that has led to the discovery of new concepts and numerous blueprints implemented in laboratories [32]. Other simulators such as *Strawberry fields* focus specifically on optimization in photonic quantum computing [33].

**Design in nanophotonics and photonic materials**    The field of optical *inverse design* focuses on the de-novo design of nano-optical components with practical features[34, 35]. Examples include on-chip particle accelerators [36], or wavelength-division multiplexers [37]. The main approach is the development of efficient PDE-solvers for Maxwell's equations, including efficient ways to compute the gradients of the vast amount of parameters, usually by a physics-inspired technique called the *adjoint method* [38, 39]. These techniques are highly computationally expensive [40] due to their physical targets. We have different physical targets, thus can apply various different approximations in the beam propagation which significantly speeds up our simulator. Interestingly, the adjoint method can be seen as a special case of auto-differentiation (which we use) [39].

**Classical optics simulators** Several open-source software tools, like *Diffractio* for light diffraction and interference simulations [41], *Finesse* for simulating gravitational wave detectors [42], and *POPPY*, developed as a part of the simulation package of the James Webb Telescope [43], facilitate classical optics phenomena simulations. There are also specialized resources like those focusing on the design of Laguerre-Gaussian mode sorters utilizing multi-plane light conversion (MPLC) methods [44]. While these software offer optics simulation capabilities, XLUMINA uniquely integrates simulation with AI-driven automated design powered with JAX's autodiff and just-in-time (jit) compilation capabilities.

## 2 Software workflow and performance

XLUMINA allows for the simulation of classical optics hardware configurations and enables the optimization and automated discovery of new setup designs. The software is developed using JAX [45], which provides an advantage of heightened computational speed while seamlessly integrating the auto-differentiation framework [23]. It is important to remark that our approach is not restricted to run on CPU (as NumPy-based softwares do): due to JAX-integrated functionalities, by default runs on GPU or TPU if available, otherwise automatically falls back to CPU.

The first benchmark is to rediscover highly impactful microscopy strategies, such as STED microscopy [10] or the sharp focus of a radially polarized light beam [24], as each of these incorporate different ideas or physical properties of light. To that end, the algorithm is equipped with an optics simulator, which contains a diverse set of optical manipulation, interaction, and measurement technologies. The simulator enables, among many other features, to define light sources (of any wavelength and power), phase masks (i.e., spatial light modulators, SLMs), polarizers, variable retarders (e.g., liquid crystal displays, LCDs), diffraction gratings, and high NA lenses to replicate strong focusing conditions. Light propagation and diffraction is simulated by two methods, each available for both scalar and vectorial regimes: the fast-Fourier-transform (FFT) based numerical integration of the Rayleigh-Sommerfeld (RS) diffraction integral [46, 47] and the Chirped z-transform (CZT) [48]. The CZT is an accelerated version of the RS algorithm, which allows for arbitrary selection and sampling of the region of interest. Some functionalities of XLUMINA's optics simulator (e.g., optical propagation algorithms, planar lens or amplitude masks) are inspired in an open-source NumPy-based Python module for diffraction and interferometry simulation, *Diffractio* [41], although we have rewritten and modified these approaches to combine them with JAX's just-in-time (jit) functionality. On top of that, we developed completely new functions (e.g., LCDs or propagation through high NA objective lens with CZT methods, to name a few) which significantly expand the software capabilities. The most important hardware addition on the optical simulator are the SLMs, each pixel of which possesses an independent (and variable) phase value. They serve as a universal approximation for phase masks (including lenses) and offer a computational advantage: given a specific pixel resolution, they allow for unrestricted phase design selection. Such flexibility is crucial during the parameter space exploration, as it allows the software to autonomously probe all potential solutions. In addition, we defined under the name of *super-SLM* (sSLM) a hardware-box-type which consists of two SLMs, each one independently imprinting a phase mask on the horizontal and vertical orthogonal polarization components of the field.

To include the automated discovery feature, XLUMINA's optical simulator and optimizer are tied together by the loss function. The software's workflow is depicted in Fig. 1. We start by feeding the system an initial random set of optical parameters, which shape the hardware design on a virtual optical table. The performance of the virtual experiment is computed by the simulator, which leads to detected light (e.g., captured images at the camera). From those simulated outputs, the objective function (for instance, the spot size), is computed. To improve the metric of the cost function, the optimizer adjusts the optical parameters in the initial virtual setup and the cycle is repeated. The whole process is a back-and-forth between the simulator and the optimizer, refining the setup until a convergence is observed.

The automated discovery tool is designed to explore the vast parameter space encompassing all possible optical designs. A direct outcome of running individual optical simulations during each optimization iteration is the considerable computational expense. Thus, it is essential to reduce the computation time by maximizing the speed of optical simulation functions. By strategically leveraging the JAX's jit functionality, we optimize already existing propagation algorithms to mitigate this computational constraint. Thus, we evaluate the performance of our optimized functions against

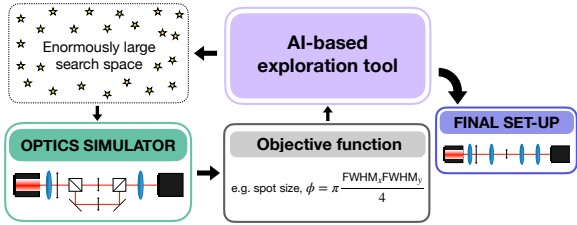

Figure 1: Workflow of XLUMINA, demonstrating the integrated feedback between the AI discovery tool and the optics simulator.

their counterparts in *Diffractio* by propagating a Gaussian beam within a computational window sized at $2048 \times 2048$. The average run-time for both *Diffractio* and our approach is shown in Figure 2a. Generally, our methods significantly enhance computational speeds for simulating light diffraction and propagation. For instance, we observe a speedup of roughly a factor of 2 for RS and VCZT and about 2.5 for VRS using the CPU. CZT has less significant speedup, but there is still a 0.5-second improvement. With GPU utilization, the speed increases by up to three orders of magnitude.

a)

|  | CPU | | | |
|---|---|---|---|---|
|  | RS | CZT | VRS | VCZT |
| *Diffractio* | 4.14 | 1.91 | 12.33 | 6.17 |
| *Our approach* | **2.39** | **1.39** | **5.22** | **4.04** |

|  | GPU | | | |
|---|---|---|---|---|
|  | RS | CZT | VRS | VCZT |
| *Diffractio* | / | / | / | / |
| *Our approach* | **0.006** | **0.027** | **0.151** | **0.075** |

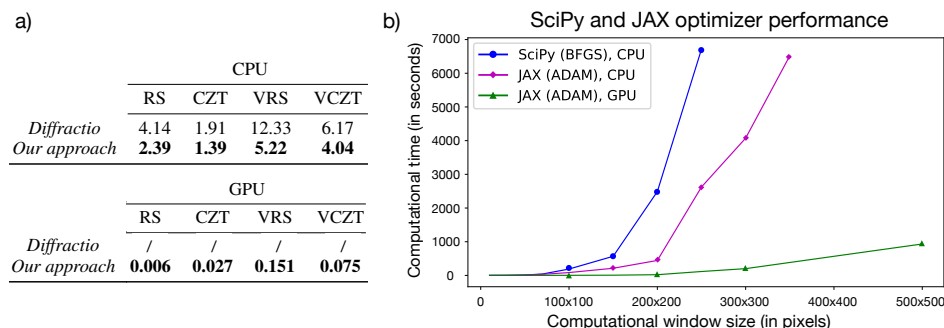

Figure 2: (a) Average execution time (in seconds) over 5 runs for scalar and vectorial field propagation using Rayleigh-Sommerfeld (RS, VRS) and Chirped z-transform (CZT, VCZT) in *Diffractio* and *our approach*. GPU times reflect the second run with pre-compiled jitted functions. (b) Convergence time (in seconds) of BFGS and ADAM optimizers, on CPU and GPU, for computational windows of sizes up to $500 \times 500$ pixels. The superior efficiency of the optimizer on GPU allows for highly efficient optimizations in the large computational windows we use (up to $2048 \times 2048$).

When it comes to the nature of the optimizer, it can be either direct (gradient-based) or deep learning-based (surrogate models or deep generative models, e.g., variational autoencoders [49]). In this work, we adopt a gradient-based strategy, where the experimental setup's parameters are adjusted iteratively in the steepest descent direction. To chose the optimizer, we evaluate the convergence time of two gradient-descent techniques: the Broyden-Fletcher-Goldfarb-Shanno (BFGS) algorithm, which numerically computes gradients and higher-order derivative approximations, and the adaptive moment estimation (ADAM), an instance of the stochastic-gradient-descent (SGD) method. While BFGS is part of the open-source SciPy Python library and operates on the CPU, ADAM is integrated within the JAX library and runs in both CPU and GPU. Taking advantage of the JAX's built-in autodiff framework, the gradients of the loss function are computed analytically. Combined with the jit functionality, this approach enables the optimizer to efficiently construct an internal gradient function, thus considerably reducing computational time per iteration. For the evaluation, we simulate a Gaussian beam interacting with a phase mask. The objective function is the mean squared error between the detected light and the ground truth, characterized by a Gaussian beam with a spiral phase imprinted on its wavefront. Initializing with an arbitrary phase mask configuration, we run both BFGS and ADAM optimizers over different computational windows and devices, as depicted in Fig. 2b. On the CPU, BFGS exhibits exponential scaling in convergence time, reaching about 6500 seconds for $250 \times 250$ pixel window. In contrast, ADAM demonstrates superior efficiency, reducing it to roughly 2600 seconds. GPU optimization performance is even more pronounced, reaching convergence in approximately 950 seconds for a $500 \times 500$ pixel window. Given that certain optical elements, such as phase masks, may operate at resolutions as high as $2048 \times 2048$ pixels, the resulting

search space can expand to around 8.4 million parameters if two of these elements are included. This makes the GPU-accelerated ADAM approach more appropriate for efficient experimentation. Overall, the computational performance of XLUMINA highlights its suitability for running complex simulations and optimizations with a high level of efficiency.

## 3 Results and discussion

In this section, we showcase the virtual optical designs generated by XLUMINA. As benchmarks, we aim to rediscover three foundational experiments, each one covering different areas in optics. By increasing the complexity of the description of light (from scalar to vectorial fields representation), we selected: (1) an optical telescope version, (2) polarization-based beam shaping as used in STED microscopy [10], and (3) the sharper focus for a radially polarized light beam, as detailed in Ref. [24].

**Optical telescope**  The simplified model of the telescope comprises two lenses, each one positioned a focal length apart from their respective input and output planes, $f_1$ and $f_2$, respectively, and $f_1 + f_2$ from each other. This arrangement performs optical Fourier transformations of input light with magnifications determined by the ratio $f_2/f_1$. To revisit this design with a magnification of $2x$, we encoded the virtual setup depicted in Fig. 3a, in which traditional lenses are replaced by spatial light modulators (SLMs). The parameter space includes the distances, $z_1$, $z_2$ and $z_3$ (measured in millimeters) and the phase masks (measured in radians) of the two SLMs with a resolution of $1024 \times 1024$ pixels. The training dataset is composed of $14,000$ [input, output] intensity sample pairs. Each sample consists of a Gaussian beam shaped by amplitude masks in various forms (circles, rectangles, squares and rings), with varying sizes and orientations. The corresponding output for each input is an inverted version, magnified by a factor of 2. The cost function is the mean squared error between the dataset's output and the detected intensity pattern from the virtual setup. The optimization starts with randomly initialized optical parameters. We select training examples in batches of 10 and evaluate the current setup response and its loss value. The average loss over the batch guides the update of the optical parameters, repeating this cycle until convergence is reached.

The obtained results are displayed in Fig. 3b. The solution depicts lens-like quadratic phases in both SLMs. Notably, the reference model traditionally uses two lenses set at specific distances, yet the identified distances don't fulfill such relation. This suggests that phase mask of SLM#1 might be compensating for this deviation. The challenge ahead lies in effectively balancing the optimization of millions of phase parameters against a singular distance parameter. On the other hand, we believe that the more precise solution for SLM#2 highlights its critical role in imaging. The triangle-shaped amplitude mask shown in Fig. 3c, not included in the training data, shows the optical setup can invert but not sufficiently magnify the input shape.

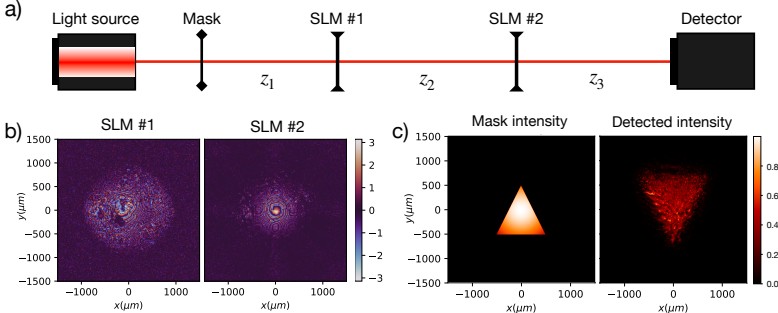

Figure 3: (a) Virtual telescope arrangement with original lenses replaced by two spatial light modulators (SLMs). Optimized distances correspond to $z_1 = 0.74$ mm, $z_2 = 2.58$ mm and $z_3 = 4.14$ mm. (b) Phase mask solutions for SLM#1 and SLM#2. (c) Input and detected intensity patterns for the identified optical design.

**STED microscopy**  STED microscopy [10] is based on excitation and spatially targeted depletion of fluorophores. In order to achieve this, a Gaussian-shaped excitation beam and a doughnut-shaped

depletion beam are concentrically overlapped. The depletion beam has zero intensity in the center, where the excitation beam has its maximum. Fluorophores that are not in the center of the beams are forced to emit at the wavelength of the depletion beam. Their emission is spectrally filtered out. Only fluorophores in the center of the beams are allowed to fluoresce normally, and only their emission is ultimately detected. This effectively reduces the area of normal fluorescence, which leads to super-resolution imaging.

In order to generate a doughnut-shaped beam a spiral phase is imprinted into the wavefront of a Gaussian beam. To revisit this principle, we virtually construct a simplified version of a STED-type setup as depicted in Fig. 4a. It consists of two light sources generating Gaussian beams corresponding to the depletion and excitation beams with wavelengths of 650 nm and 532 nm, respectively. The excitation and depletion beams are linearly polarized in orthogonal directions. Within the depletion beam's optical path, we place an SLM of $2048 \times 2048$ resolution and a computational pixel size of $1.95 \mu m$. After propagating some set distance, a high numerical aperture (NA) objective lens focuses both beams onto the detector screen. To simulate the basic concept of stimulated emission with neither time dependency nor fluorophores in the focal plane, we perform a subtraction of the intensity of the excitation and depletion beams, which results in the effective fluorescence that would ultimately be detected (negative values are set to zero, resembling a filter that removes residual depletion intensity). In this instance, the parameter space is defined by the SLM. The loss function is calculated as the inverse of the normalized intensity, $I_{det}$, over the detector. Only pixels with $I_{det} > \varepsilon \cdot \max(I_{det})$, where $0 < \varepsilon < 1$, are considered, with all others set to 0. In particular,

$$\mathcal{L} = \frac{1}{\sum_{i,j}^{N} I_\varepsilon(i,j)}, \tag{1}$$

where $N$ is the camera's total pixel count and $I_\varepsilon$ is the intensity pattern for a given threshold $\varepsilon$. Thus, minimizing $\mathcal{L}$ aims to maximize the generation of high intensity beams. For this particular instance, the detected intensity corresponds to the radial component, $|E_x|^2 + |E_y|^2$, of the effective beam.

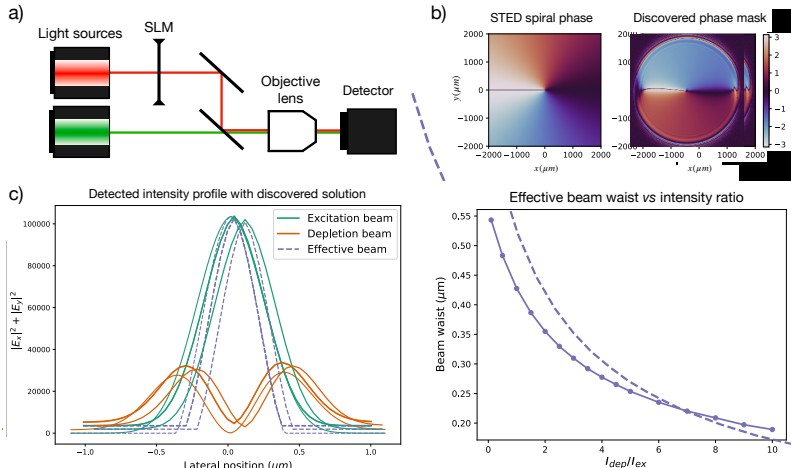

Figure 4: (a) Virtual STED-type setup. A 0.9 NA objective lens focuses both light beams into the detector screen of $0.05 \mu m$ pixel size. (b) STED spiral phase (Hell, S. and Wichmann, J., 1994) and discovered phase mask. (c) Radial intensity profile in vertical beam section: excitation (green), depletion (orange), and super-resolution effective STED beam (dotted blue). Lateral position indicates lateral distance from the optical axis. (d) Effective beam waist (in $\mu$m) as a function of depletion and excitation intensity ratio ($I_{dep}/I_{ex}$).

In Fig. 4b, we present the STED spiral phase mask [10] and the identified solution for $\varepsilon = 0.7$. From a random initial phase mask in the SLM, the system converged into a pattern alike to the spiral phase. While the spiral phase mask features a consistent and gradual phase variation across the spiral, this progression is not as evident in the discovered solution. Furthermore, we would like to emphasize the remarkably low noise contribution on the identified phase pattern. Other solutions presented noisy phase patterns which failed to achieve the essential doughnut-shaped depletion beam. Real-world

STED setups demand almost perfect phase patterns; even the minor misalignment can compromise the super-resolution STED phenomena. Remarkably, without prior knowledge, our system detected this sensitivity, converging towards a smooth phase pattern. To highlight the doughnut shape of the depletion beam, we computed the vertical cross-section of the focused intensity patterns for both excitation and depletion beams (green and orange lines in Fig. 4c, respectively) and the effective beam resulting from stimulated emission (dotted blue line in Fig. 4c). The behavior across the horizontal axis yields similar features. Notably, despite both the excitation and depletion beams being diffraction-limited, the effective response is sub-diffraction. To expand on this AI-discovered solution, we systematically changed the intensity of the depletion beam relative to the excitation beam. Indeed, we observed the expected inverse square root scaling of the effective beam diameter relative to the intensity ratio of depletion and excitation beam (4d). Such outcomes accentuate the success of our AI-driven exploration tool in identifying crucial components intrinsic to STED microscopy.

**Sharper focus for a radially polarized light beam**  The final benchmark focuses on the generation of an ultra-sharp focus for a radially polarized beam, a feature that breaks the diffraction limit in the longitudinal direction as demonstrated by R. Dorn, S. Quabis and G. Leuchs in Ref. [24]. This super-resolution is achieved when a radially polarized beam is tightly focused using a high NA objective lens [50]. Importantly, by rotating the input polarization by $90^o$ it is possible to switch from radially to azimuthally polarized beam while maintaining the same doughnut-shaped intensity distribution. In the last case, however, the longitudinal electric field is zero at the optical axis [51]. To revisit this principle, we encoded the virtual setup depicted in Fig. 5a. The light source emits a 635 nm wavelength Gaussian beam that is linearly polarized. The original optical elements are replaced by an sSLM, each component of which has a resolution of $2048 \times 2048$ pixels and a computational pixel size of $1.46\mu m$. Additionally, we place an LCD with variable phase retardance $\eta$ and orientation angle $\theta$. The beam then passes through a high NA objective lens before reaching the detector screen. Relevant data on the sSLM's phase masks, optical parameters, and the simulated spot size are showcased in Fig. 5b and Table 1. The cost function corresponds to $\mathcal{L}$ in Eq. (1), already used in the previously discussed STED setup. In this case, however, the measured intensity corresponds to the electromagnetic field's longitudinal component, $|E_z|^2$.

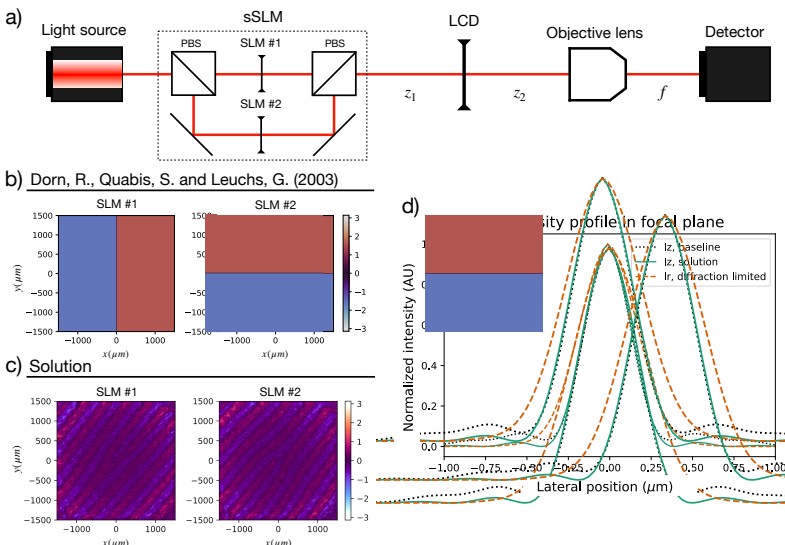

Figure 5: (a) Virtual optical setup consisting of a sSLM (super-spatial light modulator), a liquid crystal display (LCD) and an objective lens of 0.9 NA focusing the light onto the detector of $0.05\ \mu m$ pixel size; $z_1$ and $z_2$ denote for distances. (b) Phase masks for Dorn, R. et. al. (2003). (c) Discovered phase patterns for Solution #1. (d) Longitudinal intensity profile, $I_z$, for Dorn, R. et. al. (2003) and the identified Solution #1 (black dotted, and green lines, respectively) and radial intensity profile, $I_r$, of the diffraction-limited light beam (orange dotted line).

Among the obtained results we identified two interesting solutions, namely Solution #1 and #2, corresponding to $\varepsilon = 0.7$ and $0.5$, respectively. The obtained phase patterns for Solution #1, depicted

in Fig. 5c, resemble forked gratings with integer forked dislocations of $p = 1$ [52]. Such gratings are known for producing beams with phase singularities, like doughnut-shaped beams, which play an important role in optical trapping and manipulation of small particles [53]. The output light beam is slightly deflected and demonstrates a radial intensity doughnut shape and a longitudinal intensity with a spot size slightly larger than the simulated for Dorn, R. et. al. [24] (see Table 1). With regards to Solution #2, the SLM phase pattern also shows a tilted forked grating of topological charge $p = 1$. However, the fringe pattern frequency of this solution is significantly higher than that of Solution #1. Due to its complexity and reduced clarity, we opted not to include it in the manuscript. The identified optical parameters are displayed in Table 1. The longitudinal intensity profiles (assuming all beams on-axis) of Dorn, R. et. al. (2003) and Solution #1 are depicted in Fig. 5e (represented by dotted black and green respectively). For comparison, we also feature the radial intensity profile of the diffraction-limited linearly polarized beam (dotted orange line in Fig. 5e). Clearly, the identified solutions surpass the diffraction limit showing similar spot sizes. Remarkably, the AI generated this solution without prior knowledge of forked grating elements or their specific use. It found an alternative way to imprint a phase singularity onto the beam and produce pronounced longitudinal components on the focal plane. Additionally, when the input polarization is rotated by $90^o$ the solution behavior is in agreement with Dorn, R. et. al. (2003): while maintaining the same doughnut-shaped intensity distribution, the longitudinal field in the focal plane presents a zero in its center.

Table 1: Optical parameters of LCD retardance $\eta$, orientation $\theta$, propagation distances ($z_1$ and $z_2$) and simulated longitudinal spot size of Dorn, R. et. al. [24] and the identified solutions. Discovered approaches break the diffraction limit demonstrating similar spot sizes as Dorn, R. et. al (2003).

| | $\eta$ (rad) | $\theta$ (rad) | $z_1$ (mm) | $z_2$ (mm) | Spot size / $\lambda^2$ |
|---|---|---|---|---|---|
| Dorn, R. et. al. (2003) | 0 | 0 | 18 | 1000 | 0.2289 |
| Solution #1 | 2.08 | 2.56 | 15 | 19 | 0.2419 |
| Solution #2 | -1.31 | 0.80 | 34 | 20 | 0.2205 |
| Diffraction-limited | / | / | / | / | 0.27 |

**Towards large-scale discovery**    The results we've presented thus far predominantly involve optical setups characterized by a limited number of optical elements. This was crucial for our purpose to demonstrate how XLUMINA can compute and efficiently rediscover known techniques in advanced microscopy. However, our ambition extends beyond the optimization. We aim to use XLUMINA to discover new microscopy concepts. To achieve this, we will start with initial setups with a large and complex optical topology, inspired by other fields that start with highly expressive initial circuits [54, 55]. From here, XLUMINA should be able to extract much more complex solutions which humans might not have thought about yet[2]. This is our immediate next goal.

## 4   Conclusions and outlook

In this work, we present an efficient and reliable simulator for advanced optical microscopy. We demonstrate its general applicability by discovering three important and complex microscopy techniques. The simulator is developed in a modular way, and we plan to significantly expand it by adding more physical properties and features exploited in microscopy, for example, detailed coverage of frequency and time information, which might enable systems such as iSCAT [56], structured illumination microscopy [57], and localization microscopy [58]. Additionally, XLUMINA provides already the basis for an expansion to complex quantum optics microscopy techniques [59] or other quantum imaging techniques [60], as a quantum of light (i.e., a photon) is nothing else than an excitation of the modes of the electromagnetic field. Looking further into the future, one can expect that matter-wave beams (governed by Schrödinger's equation, which is closely related to the paraxial wave equation, a special case of the electromagnetic field) can be simulated in the same framework. This might allow for the AI-based design of hybrid microscopy techniques using light and complex electron-beams [61] or coherent beams of high-mass particles [62]. Ultimately, bringing so far unexplored concepts from diverse areas of physics to microscopy applications is at the heart of AI-driven discovery in this area, and we hope that this work constitutes a first step in this direction.

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
