# OpenReview forum: "XLuminA: An Auto-differentiating Discovery Framework for Super-Resolution Microscopy"
_NeurIPS.cc/2023/Workshop/AI4Science — NeurIPS2023-AI4Science Oral_

### Official Review · Reviewer_g1ky · 2023-10-06
**Exciting algorithm that is a step towards automating the discovery of novel microscopy configurations**

**Rating:** 10
**Confidence:** 4

**Review:**

## Summary

The paper deal with super-resolution microsopy where there is millions of theoretical configurations an experiment could be set up to. In practice, there is also used a range of different experimental configurations, the authors mention five (STED, PALM/F-PALM, STORM, SIM and MINFLUX). However, in order to fully explore the millions of experimental configurations, this work presents a new concept for optical microscopy, an algorithm named XLUMINA.
Here, the user need to input a desired goal. In doing so, XLUMINA can alter the experimental configuration to match this goal. The authors show that XLUMINA can rediscover three already known optical layouts but they do not show that it can discover new novel microscopy  concepts (yet).

## Evaluation of the quality

It is difficult to evaluate the quality of an algorithm were the code is not revealed. However, based on what has been written on the paper, the algorithm is well-written in JAX. I judge this by the large speed-up that XLUMINA achieves compared to Diffractio.

I will urge the authors to make a substantial supplementary information section that details all parts of the algorithm (Optics simulator, AI-based tool, etc.) including loss functions, early-stop criteria etc. and make the code open-source. If the authors does so, I think, it will be adopted by the community.

## Clarity

The paper is very well-written!

## Originality and significance

The work is original and extremely significant. Even though the authors has yet not proven that XLUMINA can find new novel microscopy concepts and it might need improvement for that, I believe that they will succeed. The step that the authors take here by showing that it can reproduce three already known optical layouts is significant.

Best of luck :)

---

### Meta-Review · Area_Chair_17BF · 2023-10-27

**Recommendation:** Accept (Oral)
**Confidence:** 5

**Metareview:**

This manuscript presents a pioneering computational framework aimed at facilitating the discovery of innovative optical hardware for super-resolution microscopy. The XLuminA framework not only promises enhancements to existing setups but also holds the potential to unearth groundbreaking concepts in microscopy. With its novelty complemented by robust experimental evidence, this work is fitting for an oral presentation. However, to fortify the introduction, I recommend referencing a selection of papers that emphasize the groundbreaking biological discoveries enabled by super-resolution microscopy, such as:

https://www.pnas.org/doi/abs/10.1073/pnas.1705043114

https://www.nature.com/articles/s41467-022-30720-x

https://www.science.org/doi/full/10.1126/science.aaw5937